# The Different Physical and Behavioural Characteristics of Zoo Mammals That Influence Their Response to Visitors

**DOI:** 10.3390/ani8080139

**Published:** 2018-08-14

**Authors:** Marina B. Queiroz, Robert J. Young

**Affiliations:** 1Conservation, Ecology and Animal Behaviour Group, Programa de Pós-Graduação em Zoologia da PUC Minas, Prédio 41, Av. Dom José Gaspar, 500, Coração Eucarístico, Belo Horizonte 30535-610, MG, Brazil; marinabonde@hotmail.com; 2CAPES Foundation Scholarship, Ministry of Education of Brazil, Brasília 70040-020, DF, Brazil; 3Acoustics Research Centre, School of Computing, Science, and Engineering, The University of Salford, The Crescent, Salford M5 4WT, UK; 4Ecosystems and Environment Research Centre, School of Environment and Life Sciences, The University of Salford, The Crescent, Salford M5 4WT, UK

**Keywords:** animal welfare, behaviour, mammals, zoo enclosures, zoo public, zoo visitors

## Abstract

**Simple Summary:**

Studies of the zoo visitor effect (changes in animal behaviour in response to the presence of the public) have had varying results: most studies have found that visitors have a negative animal welfare impact, but some studies have found no effect, or even, a positive effect on animal welfare. The problem is that most studies only consider one species at a time and meta-analyses suffer from the great variation in animal husbandry, enclosures and the public’s behaviour. Therefore, we examined 17 different mammal species responses to zoo visitors over the period of one year. The species were chosen to show a variation in physical characteristics (e.g., body weight) and habits (e.g., diurnal or nocturnal); this allows us to determine which characteristics are important in determining the variation of the response of mammals to zoo visitors. The results showed no effect of body weight, but activity cycle was very important, with diurnal animals being more affected. These results allow us to predict which mammal species will be most negatively affected by zoo visitor presence.

**Abstract:**

The factors underlying the ‘zoo visit effect’ (changes in animal behaviour/physiology in response to visitor presence) are still poorly understood, despite it being widely investigated. The present study examined the effect of zoo visitors on the behaviour of 17 different species of mammals at the Belo Horizonte Zoo, Minas Gerais, Brazil. The behaviour of the animals was recorded using scan (focal in one case) sampling with instantaneous recording of behaviour, during 12 continuous months. Data were analysed using a comparative method examining five different factors: diet, body weight, stratum occupied, activity cycle, and habitat, as well as three different visitor categories: small and quiet audience, medium size audience and medium noise, and large size and loud audience. Significant changes in the behaviour for each factor, especially increases in locomotor and resting behaviour, were observed in response to different visitor categories. The factors that most explained responses to visitor categories were habitat and activity cycle. Species from closed habitats compared to open habitats were significantly more impacted (more behaviour affected), probably, because they are, evolutionarily, less accustomed to visual contact with people. Diurnal species showed more behavioural changes than nocturnal ones, possibly, because they were being observed during their normal activity cycle. These results may help zoos identify which mammal species are most susceptible to the zoo visitor effect, and consequently, be more pro-active in the use of mitigating strategies.

## 1. Introduction

Presently, around 10 percent of the world’s human population visits a zoo every year, and, consequently, zoo animals are exposed to large numbers of visitors [1]. The presence of large number of visitors in zoos has led to a variety of studies, many of them related to the impact of visitor presence and sound on animal welfare—namely, the zoo visitor effect [2,3,4]. Studies have shown that the attitudes (behaviour) and presence of zoo visitors are associated with changes in the behaviour and physiology of captive animals [4,5,6,7,8,9].

In studies that analyse the behaviour of animals to identify the visitor effect, the expression of abnormal behaviour has often been used as an indicator of stress [4,10]. However, changes found in normal behaviours (such as resting and auto grooming) have also been used as indicators of animal stress [11]. Almost all studies have shown that the presence of visitors has some effect on animal behaviour, encouraging the expression of abnormal behaviour or changing the frequencies or durations of normal behaviours. However, Gorecki, et al. [12] found that European Souslik (*Spermophilus citellus*) is active and expresses normal behaviour, such as eating, when exposed to human presence and human noise. Furthermore, a study by O’Donovan, et al. [13] found that cheetahs’ (*Acinonyx jubatus*) behaviours are not affected by visitor presence.

Some studies have shown that the presence of visitors can be enriching for the animals in captivity; for example, animals which are fed by visitors express higher activity levels and disperse through their enclosure to gain an advantage in the search for food thrown by visitors (Fa, 1989, as cited in [2]). In contrast, most studies show that the presence of the public in zoos has a negative impact on animal behaviour, such as reducing social interactions (grooming, for example [14]). The presence of visitors caused an increase in abnormal and aggressive behaviours in *Macaca silenus* [10], an increase in agonistic behaviours in three species of primates (*Saquinus oedipus*, *Cercopithecus diana* and *Lemur catta*), and an increase in abnormal behaviours and activity in *Mandrillus leucohpaeus* [14]; furthermore, jaguars (*Panthera onca*) suffered prolonged stress and agitation from the presence of the public [15]. Several studies with gorillas (*Gorilla gorilla*) revealed that they are more agitated in a negative way in the presence of visitors and that they are more aggressive [11,16,17]. Mallapur and Chellam [18] showed that Indian leopards (*Panthera pardus*) rest more in visitor presence and that they are more active in the absence of public. The same animals expressed more abnormal behaviours when they were observed by visitors.

In most of the aforementioned studies, behavioural observations were performed over a few months with a high daily sampling effort. This form of data collection has proven to be effective. However, Margulis and Westhus [19] demonstrated that data collected at regular intervals for long periods of time have the same efficiency; and in studies that use analysis of common behaviours of animals, the possibility of statistical error is low. Based on that, this study was conducted with constant data collection over a period of one year.

Most studies in zoos have been conducted with primates or only one species as research subject [4]. These studies are limited because they cannot undertake comparisons of factors, which can affect how different species respond to zoo visitors. Research that simultaneously analyses different species—the comparative method—are able to explain the differences between species’ responses. They can indicate which groups of species should be examined more carefully in terms of their welfare, for example [20].

The physical and behavioural characteristics of different species may be predictive of their response to the visitor effect: herbivores may be expected to have a stronger response than carnivores due to their evolutionary history as being a prey species; larger species may feel less intimidated by humans; diurnal species may be more affected than nocturnal as disturbance would happen during their normal activity cycle; closed habitat species would be expected to have a stronger response than species from open habitats, if they cannot fully hide; and arboreal species may feel more secure than terrestrial species, if they have appropriate substrates. These predictions have not been tested before in zoo visitor studies.

Based on the previous research presented above, we intend to overcome some presented limitations. Therefore, we tested the hypothesis that species differences in diet, activity cycle, body weight, habitat, and stratum used will affect how mammal species behaviourally respond to zoo visitors. We predict that there will be differences: between carnivorous, herbivorous, and omnivorous species; between light and heavy animals; between terrestrial and arboreal animals; between animals from open and closed habitats; and between animals with diurnal and nocturnal habits, as expressed above. We did this by observing the behavioural response of 17 different species of mammals to the presence of different group sizes of zoo visitors.

## 2. Materials and Methods

### 2.1. Study Site

The study was conducted at the Belo Horizonte Zoo (19°51′20.38″ S; 44°00′19.95″ W), which is located in Belo Horizonte, Minas Gerais, Brazil.

### 2.2. Animals

Seventeen species were chosen to represent different types of mammals from the zoo’s collection, varying according to the following factors: diet (herbivorous, carnivorous, or omnivorous), body weight (divided into four categories), activity cycle (diurnal or nocturnal), habitat (closed—animals from habitats such as forests, or open—animals from habitats such as savannahs), and stratum use (terrestrial, arboreal, or scansorial) (Table 1). All species chosen for the study were housed in the zoo for at least a year and they presented no evidence of habituation to the presence of visitors. Each species was observed in a different enclosure, all areas provided some kind of shelter to the animals.

### 2.3. Data Collection

Data collection was performed on days the zoo was open to the public, in the morning, Tuesday through to Saturday, during 12 continuous months, from April to March (43 days in total). To this end, we created a list of observable behaviours that were possible to observe in all our species (Table 2). During the observations, the researcher noted the animal’s behaviour and the characteristics of the public—number and agitation, as described below. Previously, we made an analysis of the different types of public. The number of people was divided into three groups: small, less than 10 people; medium, between 10 and 20 people; and large, more than 20 people. The agitation was measured on a subjective scale, following the researcher perception of noise, classified as quiet, medium, and loud noise. It was possible to observe that in Belo Horizonte Zoo existed three types of public: small and quiet crowd, medium with medium noise crowd, and large and loud crowd. For data collection, we used Scan (for 15 species) and Focal (for two species, which was housed on their own) sampling with instantaneous recording. Individuals of each species were observed four times on each day of data collection. Species were observed in a prescribed sequence following a Latin square design to avoid time-of-day bias in data collection. Firstly, all species were observed once, and then the sequence was followed again, and so on, for a total of four observations. From the moment the researcher arrived in front of the first enclosure, data were immediately collected, then the researcher moved to the next enclosure, and so on. Each species was observed 172 times. One researcher collected all the data.

### 2.4. Statistical Analysis

For data analysis, the species were classified into five different groups with subcategories: diet (carnivorous, herbivorous, or omnivorous), body weight (1–10 kg, 11–50 kg, 51–99 kg, or 100–999 kg), stratum use (terrestrial, arboreal, or scansorial), open or closed habitat, and activity cycle (diurnal or nocturnal). The public was divided into three different groups: Small and quiet audience; Medium audience and medium noise; and Large and loud audience. More than 95% of our audiences fell into these three categories. Other possibilities, such as a small and loud audience or a large and quiet audience, were not considered because they occurred in less than 5% of all observations. The behaviour data were converted into percentages for each month per species, factor, and visitor categories.

The collected data did not meet the requirements for a factorial analysis of variance, which would allow the investigation of the interaction between the factors and the public categories; therefore, the General Linear Model (GLM) tests were chosen. The requirements for the test were verified and statistically accepted. GLM models with Gaussian family were used to determine the existence of significant changes in animals’ behaviours by the aforementioned factors and by visitor presence. Five groups of GLM tests were run in which the behavioural data were used as response variable and the factors and visitor categories were used as explanatory variables. Interaction tests were used to investigate the influence of the public on the analysed factors. All tests were performed using RStudio software (1.1.456, RStudio, Boston, MA, USA) [34].

## 3. Results

Each species was observed 172 times during the year of the data collection. Descriptive data can be found in Table 3.

### 3.1. Habitat

For this factor, the species from open habitat were used as a reference in the GLM tests. Species from closed habitat presented an increase in the frequency of eating in the presence of large public groups and an increase in positive interactions in the presence of medium public category compared to open habitat species. On the contrary, the alert behaviour presented a decrease expressed by the closed habitat species, in comparison to open habitat ones, in the presence of medium and large public groups (Table 4).

### 3.2. Activity Cycles

For the activity cycles, in the GLM tests, the diurnal species were used as a reference for the presented results. In comparison to diurnal species, the nocturnal ones ate less, were more alert, and moved less away from the visitors when large public groups were around. Moreover, still in comparison to diurnal animals, the nocturnal species expressed fewer positive interactions and move less in the direction of visitors when a medium public category was present (Table 5).

### 3.3. Diet

The diet factor results will be presented with the herbivorous animals used as a reference and all the following described results will be presented in comparison to this factor category. The carnivore species moved more, rested less, expressed less positive social interactions, and approached more the visitors when medium visitor groups were around, and were more alert and moved more away from visitors when large visitor groups were present. The omnivorous mammals ate more and rested more in the presence of large visitor groups. Interestingly, the positive interactions behaviour were affected in different ways in the omnivorous animals. The animals classified in this factor decrease the positive interaction when medium visitor group was present and interacted more when large visitor groups were around (Table 6).

### 3.4. Stratum Use

For the interpretation of stratum results, the terrestrial animals were used as a reference and all the following described results will be presented in comparison to this factor category. Arboreal and scansorial species presented the same response pattern in the presence of large visitor groups eating more, drinking less, and resting less. In addition, the alert behaviour was less expressed by arboreal and scansorial species in the presence of medium visitor category and by the scansorial species in the presence of large groups. Arboreal animals interacted more when medium size visitor groups were present and less and large groups were around. And the scansorial species moved more away from visitors and was more non-visible in the presence of large visitor groups (Table 7).

### 3.5. Body Weight

This factor did not produce any significant differences in behaviour in interactions with audience categories.

## 4. Discussion

For the first time, it was possible to consider the different responses of mammals to the zoo visitor effect as influenced by different physical and behavioural characteristics of species. Changes in individual behaviours can be interpreted in different ways. For example, an animal that rests more in the presence of visitors can be seen as an animal that is comfortable in the environment, even with public around, or could be seen as an animal that is resting more to avoid the visitor’s presence. Based on the literature already published on different species and on the conclusions made by authors we will discuss this in our results below. However, it is important to state that this is not a definitive discussion on the subject. This study was produced with the intent of being a first attempt at discussing zoo mammals’ response to visitors based on their physical and behavioural characteristics. 

From the results, it is possible to see that carnivorous animals were less affected than herbivorous or omnivorous, terrestrial animals were more influenced by visitors than arboreal ones, closed habitat animals were more affected by visitors than open habitat species, and diurnal species were more affected by visitors than nocturnal animals. This research highlights the importance of previously overlooked factors in understanding which species are most impacted by the zoo visitor effect. For example, we can now begin the prediction that a species from closed habitat, also diurnal, terrestrial, herbivore, or omnivore would be most affected by the zoo visitor effect (e.g., deer species independent of body size).

The results showed that the presence of visitors resulted in changes in mammal behaviour: for example, a significant increase in the frequency of resting and locomotion behaviours. This evidence agrees with Hosey [35] who states that the public in zoos directly affects animals’ activity level, which could be associated with stress. We noticed that for almost all of the factors analysed, locomotor behaviour was most commonly affected; in agreement with another study, which found that large audiences increased the locomotor activity of 12 primate species by 25% [2]. This issue should be highlighted since an increase in locomotor behaviour could be a response to the animal’s necessity developing into abnormal behaviour such as stereotypic pacing [36]. A study by Mallapur, Sinha and Waran [10] concludes that, in the presence of visitors, levels of pacing for *Macaca silenus* increased considerably. Another behaviour that also showed significant changes in virtually all factors was resting; confirming the observation by Wells [11] that high public density encourages an increase in grooming by gorillas (*G. gorilla*).

### 4.1. Habitat

Species from closed habitats were more affected than species from open habitats. We can infer from this that closed habitat species are not accustomed to contact with people, since they usually live in places where they can hide such as tropical forests, and because of this, they felt more threatened by the presence of visitors. For example, Sellinger and Ha [15] conclude that, at higher density and agitation of the public, the stress in jaguars (*Panthera onca*) is prolonged. Another example is the species *Macaca silenus*, which lives in tropical forests, rarely, interacts with humans in their natural habitat. For this species, the contact with visitors in zoos can be a real source of stress [10]. However, this situation may change if adequate opportunities to use their cryptic habits were provided. Stress levels of captive leopard cats (*Felis bengalensis*) reduced significantly when opportunities to camouflage themselves and/or use hiding places were provided [37]. Open habitat species may be more habituated to people (they often live in open environments such as savannahs) and, therefore, presented fewer behavioural changes compared to closed habitat species.

### 4.2. Activity Cycle

Diurnal species had more of their behavioural repertoire affected when compared to nocturnal species by zoo visitors. Observations were made during the day, the hours of visitation at the zoo, coinciding with the time activity of the diurnal species. This may have caused the greatest impact for those animals who may have felt uncomfortable and invaded by the presence of visitors. Nocturnal animals usually become inactive during the day, and possibly this was the reason they showed fewer variations in behaviours by the presence of visitors that can be a cause of concern, as they were most of the time sleeping or hiding. More studies are needed of this factor such as observations of nocturnal species at night in captivity.

### 4.3. Diet

Carnivorous animals being predators and, therefore, naturally more aggressive possibly felt less threatened by the presence of visitors. Our results showed this trend because carnivores had fewer behaviours that could be considered a cause of concern affected by the presence of visitors compared to herbivores or omnivores. Herbivores and omnivores are more often prey species for other animals and may perceive human visitors at their enclosures as predators due to evolutionary history [38]. The lack of other zoo visitor studies comparing the behavioural repertoire of animals with different dietary habits hinders a deeper understanding of our results.

### 4.4. Stratum Used

Arboreal species virtually showed no important behavioural changes in response to visitors. This could be explained by the fact that the composition of the enclosure with trees and shelter provided the animals with control over of their exposure to the public [35]. However, only two species in this study had arboreal habit, which could have influenced the result. Terrestrial and scansorial animals, probably by being more exposed in their enclosures, lost some control over their exposition to the public. These animals showed a significantly greater impact on their behavioural repertoire. This disagrees with Hosey [2] who indicated a tendency for arboreal animals to be more affected by the presence of people than terrestrial animals. This difference may be due to the fact that arboreal animals in this study could maintain a minimum distance from the public because of the provision of standoff barriers (around one meter). Some zoos do not have this kind of barrier that separates the public from the edge of the animals’ enclosure.

## 5. Conclusions

This study identified which mammal species characteristics are most affected by the visitor effect and which factors are most associated with a negative response to this effect, for example, being a species from a closed habitat or a herbivorous animal. Therefore, we propose a predictive model showing the characteristics of species that leads to most or least concern relating to visitors in a zoo environment (Figure 1). As an example, a deer fit in the model as an animal from open area, with diurnal, herbivorous, and terrestrial habits. Thus, it is a species in the most concern category and could be focus of more attention in the zoo when thinking about the visitor effect. On the other hand, chimpanzees, that are animals from closed habitats, omnivorous, scansorial and diurnal, are of least concern to the zoos in regards to the visitor effect. The use of this tool could help zoos better design enclosures and management protocols for their species, or even create environment enrichment activities for animals and interventions of environmental education with zoo visitors.

As mentioned previously, the model we have produced to predict zoo mammals’ response to visitor presence is a first iteration and should not be treated as definitive, especially as our sample size is small. In all areas of research when producing predictive model, it is necessary to have a first iteration, which with other studies will be modified so that future iterations will be more accurate in their predictions. We hope that this model will now be subject to testing and modification by other researchers. Comparative studies such as this are important as they illuminate the different underlying factors that can affect the response being measured. However, for a complete investigation concerning the zoo visitors’ effect, many other possibilities can be approached. For example, it would be necessary that other groups of animals besides mammals be studied such as birds and reptiles. Another proposal is to investigate nocturnal animals during their time of activity to thereby make an appropriate analysis of their welfare and responses to the public. Some zoos offer nocturnal visits where this kind of study could take place or species in zoos with nocturnal houses could be studied.

During the study, it was not possible to analyse the behaviour of animals in absence of public, because even with the presence of the researcher, there was a person present. A study that could be done is to compare differences in behavioural repertoires of animals in presence of visitors and total absence of them. For this, the proposal is to use video cameras located in the enclosures of the animals to record behaviours without an audience presence. For future research, quantified sound measurements with specific equipment should be made to investigate the quantitative difference between quiet and loud visitors.

Some previous studies analysed the physiological effect of zoo visitor on animals, showing that cortisol levels can be directly related to levels of stress [5,39,40,41]. Special attention is necessary when choosing the type of sample that will be used for corticoid measurements. Some methods (corticoid measurements from blood, saliva, milk) needs intense manipulation of the animals, which can imply in a source of stress to the animal. Therefore, the most common and non-invasive method to assess the physiology of animal welfare is the use of faecal samples [42,43]. This type of physiological measurements could be done in futures studies to provide a complement of the behaviours analysis in relation to visitors’ presence.

## Figures and Tables

**Figure 1 animals-08-00139-f001:**
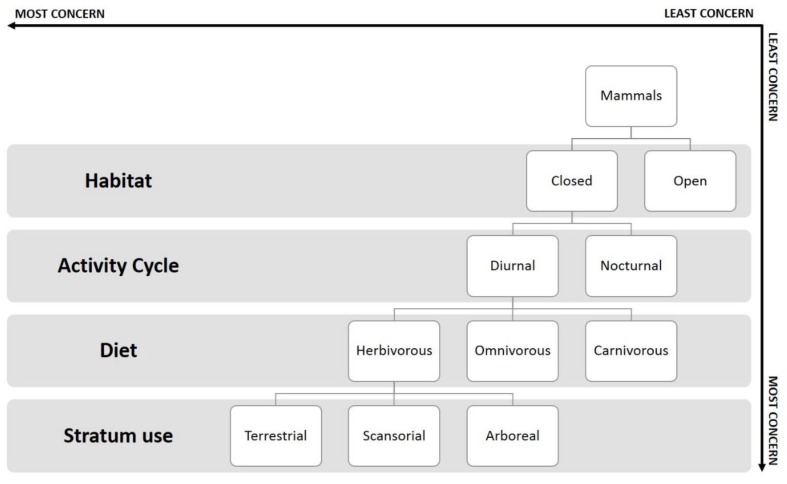
First iteration of a predictive model for zoo mammals’ response to visitors’ presence. Species located on the top-right of the diagram could be considered of least concern regarding the zoo visitors’ effect. Species located on the bottom-left could be considered of most concern regarding the zoo visitors’ effect.

**Table 1 animals-08-00139-t001:** Mammals species observed, number of animals, factors that characterise the animals ^1^ and brief description of their enclosures at the Belo Horizonte Zoo, Minas Gerais, Brazil.

Mammals	Scientific Name	♂	♀	Diet	Stratum	Activity	Weight	Habitat	Enclosure	Area
Bush dog	*Speothos venaticus*	2	2	Carnivorous	Terrestrial	Diurnal	6 kg	Closed	Wall with protection grid	260 m^2^
Chimpanzee	*Pan troglodytes*	2	2	Omnivorous	Scansorial	Diurnal	53 kg	Closed	Moat with protection grid	1256.64 m^2^
Coatis	*Nasua nasua*	0	1	Omnivorous	Scansorial	Diurnal	5 kg	Closed	Wall without protection grid	180 m^2^
Common eland	*Taurotragus oryx*	1	3	Herbivorous	Terrestrial	Diurnal	640 kg	Open	Wall with protection grid	1560 m^2^
Fallow deer	*Dama dama*	3	2	Herbivorous	Terrestrial	Diurnal	73 kg	Open	Wall with protection grid	800 m^2^
Gorilla	*Gorilla gorilla*	1	0	Herbivorous	Scansorial	Diurnal	200 kg	Closed	Wall with protection grid	2000 m^2^
Grison	*Galictis vittata*	2	2	Carnivorous	Scansorial	Diurnal	3 kg	Open	Wall without protection grid	104 m^2^
Howler monkey	*Alouatta fusca*	2	1	Herbivorous	Arboreal	Diurnal	6 kg	Closed	Cage with protection grid	40.2 m^2^
Lion	*Panthera leo*	1	1	Carnivorous	Terrestrial	Nocturnal	225 kg	Open	Moat with protection grid	1256.64 m^2^
Mandrill	*Mandrillus sphinx*	1	1	Omnivorous	Scansorial	Diurnal	30 kg	Closed	Moat with protection grid	1256.64 m^2^
Maned wolf	*Chrysocyon brachyurus*	0	1	Omnivorous	Terrestrial	Nocturnal	23 kg	Closed	Wall with protection grid	576 m^2^
Marsh deer	*Blastocerus dichotomus*	1	1	Herbivorous	Terrestrial	Diurnal	150 kg	Open	Wall with protection grid	1080 m^2^
Ocelot	*Leopardus pardalis*	3	0	Carnivorous	Scansorial	Nocturnal	13 kg	Closed	Moat with protection grid	1256.64 m^2^
Puma	*Puma concolor*	3	0	Carnivorous	Terrestrial	Nocturnal	63 kg	Closed	Moat with protection grid	1256.64 m^2^
Red deer	*Cervus elaphus*	1	3	Herbivorous	Terrestrial	Diurnal	295 kg	Open	Wall with protection grid	1040 m^2^
Tiger	*Panthera tigris*	1	1	Carnivorous	Terrestrial	Nocturnal	330 kg	Closed	Moat with protection grid	1256.64 m^2^
Titi monkey	*Callicebus nigrifrons*	3	1	Herbivorous	Arboreal	Diurnal	1 kg	Closed	Cage with protection grid	32 m^2^

^1^ Data obtained from the [21].

**Table 2 animals-08-00139-t002:** List of behaviours observed and recorded during a study into the visitor effect on mammals at the Belo Horizonte Zoo, Minas Gerais, Brazil.

Behavioural Categories	Initials	Description of Behaviours
Locomotion	Loc	The animal moves through the enclosure, but not specifically in the direction or away from the public.
Eating	Eat	The animal feeds.
Drinking	Dri	The animal drinks water.
Resting	Res	The animal is still but can be performing some activity.
Sleeping	Sle	The animal is still or lying down with its eyes closed and apparently asleep.
Vigilance	Vig	The animal is attentive to its environment, with its head held high.
Vocalisation	Voc	The animal vocalises.
Positive interaction	PI	Animals from the same group interact positively in a non-agonistic way (grooming, for example).
Negative interaction	NI	Animals from the same group interact negatively in an agonistic way (aggression, for example).
Abnormal behaviour	AB	Repetitive and non-wild type behaviour.
Depart	Dep	The animal hides or moves away from the public.
Approach	App	The animal walks toward the public.
Others	Oth	Other behaviours not listed.
Not visible	NV	The animal is not visible in its enclosure.

Species-specific information used as a base for the current ethogram can be found in: (Bush dog—[22,23]; Chimpanzee—[24]; Coati—[25]; Common eland, Fallow deer, Marsh deer, and Red deer—[26]; Gorilla—[27]; Grison—[28]; Howler monkey—[29]; Lion, Ocelot, Puma, and Tiger—[30]; Mandrill—[31]; Maned wolf—[32]; Titi monkey—[33]).

**Table 3 animals-08-00139-t003:** Behaviours registers by species during a study into the visitor effect on mammals at the Belo Horizonte Zoo, Minas Gerais, Brazil (Behaviours initials according to Table 2).

Species	Loc	Eat	Dri	Res	Sle	Vig	Voc	PI	NI	AB	Dep	App	Oth	NV
Bush dog	62	0	1	10	11	2	2	0	0	0	22	15	1	75
Chimpanzee	46	43	1	113	90	3	1	42	0	0	12	1	19	64
Coatis	47	10	0	12	114	16	0	0	0	0	4	2	0	0
Common eland	14	39	1	53	19	20	0	0	0	0	1	0	0	88
Fallow deer	30	14	3	92	81	13	0	2	0	0	24	0	0	63
Gorilla	5	12	0	23	25	0	0	0	0	0	65	0	0	33
Grison	189	8	2	23	4	22	1	6	0	0	17	12	1	56
Howler monkey	49	6	0	114	109	10	5	43	0	0	1	0	0	3
Lion	9	1	1	52	133	31	3	6	0	1	16	0	0	8
Mandrill	45	22	0	48	35	2	0	6	0	0	13	0	0	51
Maned wolf	27	4	0	19	14	5	0	0	0	0	69	2	1	40
Marsh deer	10	8	0	86	100	15	0	0	0	0	2	0	0	22
Ocelot	21	1	2	37	162	4	1	0	0	0	13	0	0	25
Puma	21	0	2	47	175	24	2	0	0	0	7	0	0	36
Red deer	21	37	5	104	75	32	5	8	0	0	3	0	0	66
Tiger	67	0	4	47	156	19	0	0	0	0	5	0	2	5
Titi monkey	87	24	0	129	35	4	0	97	0	0	10	4	0	56

**Table 4 animals-08-00139-t004:** Behaviours of zoo mammals significantly affected in the two types of habitat (open and closed) and by public category (small, medium and large).

Behaviour	Independent Variables	Estimate Coefficient (±SE)	*t* Values
Eating	Closed	−0.20373 (±0.13481)	−1.511
	Medium audience	0.15482 (±0.13171)	1.175
	Large audience	−0.18607 (±0.15567)	−1.195
	Closed*Medium	0.09707 (±0.18627)	0.521
	Closed*Large	0.66771 (±0.20836)	3.205 **
Alert	Closed	−0.01591 (±0.09666)	−0.165
	Medium audience	0.68767 (±0.09444)	7.282
	Large audience	0.48056 (±0.11161)	4.306
	Closed*Medium	−0.42113 (±0.13355)	−3.153 **
	Closed*Large	−0.37792 (±0.14940)	−2.530 *
Positive interaction	Closed	0.10456 (±0.07635)	1.369
	Medium audience	0.07900 (±0.07460)	1.059
	Large audience	0.05000 (±0.08816)	0.567
	Closed*Medium	0.29701 (±0.10550)	2.815 **
	Closed*Large	−0.04187 (±0.11801)	−0.355

* *p* ≤ 0.05, ** *p* ≤ 0.01.

**Table 5 animals-08-00139-t005:** Behaviours of zoo mammals significantly affected in the two types of activity cycles (diurnal and nocturnal) and by public category (small, medium and large).

Behaviour	Independent Variables	Estimate Coefficient (±SE)	*t* Values
Eating	Nocturnal	−0.1604 (±0.1199)	−1.337
	Medium audience	0.2949 (±0.1172)	2.517
	Large audience	0.4690 (±0.1232)	3.807
	Nocturnal*Medium	−0.2180 (±0.1657)	−1.316
	Nocturnal*Large	−0.4690 (±0.1807)	−2.596 *
Alert	Nocturnal	−0.01678 (±0.10044)	−0.167
	Medium audience	0.37596 (±0.09813)	3.831
	Large audience	0.03569 (±0.10319)	0.346
	Nocturnal*Medium	0.16235 (±0.13877)	1.170
	Nocturnal*Large	0.43812 (±0.15132)	2.895 **
Positive interaction	Nocturnal	−0.11896 (±0.06439)	−1.847
	Medium audience	0.35417 (±0.06291)	5.630
	Large audience	0.03800 (±0.06616)	0.574
	Nocturnal*Medium	−0.31941 (±0.08897)	−3.590 ***
	Nocturnal*Large	−0.03800 (±0.09701)	−0.392
Move away	Nocturnal	0.3858 (±0.3066)	1.258
	Medium audience	0.1666 (±0.2996)	0.556
	Large audience	0.9833 (±0.3150)	3.122
	Nocturnal*Medium	−0.3250 (±0.4236)	−0.767
	Nocturnal*Large	−1.2639 (±0.4619)	−2.736 **
Approach	Nocturnal	3.145 × 10^−17^ (±0.01671)	0.000
	Medium audience	0.07376 (±0.01633)	4.517
	Large audience	2.621 × 10^−18^ (±0.01717)	0.000
	Nocturnal*Medium	−0.06248 (±0.02309)	−2.706 *
	Nocturnal*Large	2.338 × 10^−17^ (±0.02518)	0.000

* *p* ≤ 0.05, ** *p* ≤ 0.01, *** *p* ≤ 0.001.

**Table 6 animals-08-00139-t006:** Behaviours of zoo mammals significantly affected in the three types of diet analysed (herbivore, omnivore, and carnivore) and by public category (small, medium and large).

Behaviour	Independent Variables	Estimate Coefficient (±SE)	*t* Values
Locomotion	Omnivore	−0.09133 (±0.28091)	−0.325
	Carnivore	0.05700 (±0.28091)	0.203
	Medium audience	0.44668 (±0.27445)	1.628
	Large audience	0.33784 (±0.29795)	1.134
	Omnivore*Medium	0.26557 (±0.38814)	0.684
	Carnivore*Medium	0.77427 (±0.38814)	1.995 *
	Omnivore*Large	0.64966 (±0.42137)	1.542
	Carnivore*Large	0.32331 (±0.42965)	0.753
Eating	Omnivore	−0.24586 (±0.15225)	−1.615
	Carnivore	−0.30586 (±0.15225)	−2.009
	Medium audience	0.20098 (±0.14875)	1.351
	Large audience	0.01497 (±0.16148)	0.093
	Omnivore*Medium	0.27407 (±0.21036)	1.303
	Carnivore*Medium	−0.16338 (±0.21036)	−0.777
	Omnivore*Large	0.80449 (±0.22837)	3.523 ***
	Carnivore*Large	0.01389 (±0.23286)	0.060
Active resting	Omnivore	−0.5157 (±0.3485)	−1.480
	Carnivore	−0.4624 (±0.3485)	−1.327
	Medium audience	1.6579 (±0.3405)	4.870
	Large audience	0.3660 (±0.3696)	0.990
	Omnivore*Medium	−0.6481 (±0.4815)	−1.346
	Carnivore*Medium	−0.9889 (±0.4815)	−2.054 *
	Omnivore*Large	1.1874 (±0.5227)	2.272 *
	Carnivore*Large	0.4715 (±0.5330)	0.885
Alert	Omnivore	−0.015385 (±0.088228)	−0.174
	Carnivore	0.004615 (±0.088228)	0.052
	Medium audience	0.432998 (±0.086199)	5.023
	Large audience	0.067949 (±0.093580)	0.726
	Omnivore*Medium	−0.195483 (±0.121904)	−1.604
	Carnivore*Medium	0.053410 (±0.121904)	0.438
	Omnivore*Large	0.004968 (±0.132341)	0.038
	Carnivore*Large	0.308722 (±0.134942)	2.288 *
Positive interaction	Omnivore	−0.04713 (±0.13248)	−0.356
	Carnivore	−0.13046 (±0.13248)	−0.985
	Medium audience	0.53159 (±0.12943)	4.107
	Large audience	−0.13046 (±0.14051)	−0.928
	Omnivore*Medium	−0.38980 (±0.18305)	−2.130 *
	Carnivore*Medium	−0.48879 (±0.18305)	−2.670 **
	Omnivore*Large	0.55904 (±0.19872)	2.813 **
	Carnivore*Large	0.15644 (±0.20262)	0.772
Move away	Omnivore	0.41853 (±0.33618)	1.245
	Carnivore	−0.02535 (±0.33618)	−0.075
	Medium audience	0.20032 (±0.32845)	0.610
	Large audience	1.27631 (±0.35658)	3.579
	Omnivore*Medium	−0.18688 (±0.46450)	−0.402
	Carnivore*Medium	−0.05586 (±0.46450)	−0.120
	Omnivore*Large	−0.90842 (±0.50428)	−1.801
	Carnivore*Large	−1.17957 (±0.51418)	−2.294 *
Approach	Omnivore	−1.432 × 10^−16^ (±0.02198)	0.000
	Carnivore	−9.178 × 10^−17^ (±0.02198)	0.000
	Medium audience	0.01718 (±0.02147)	0.800
	Large audience	−3.590 × 10^−17^ (±0.02331)	0.000
	Omnivore*Medium	0.01637 (±0.03037)	0.539
	Carnivore*Medium	0.09794 (±0.03037)	3.225 **
	Omnivore*Large	2.212 × 10^−17^ (±0.03297)	0.000
	Carnivore*Large	5.933 × 10^−18^ (±0.03362)	0.000

* *p* ≤ 0.05, ** *p* ≤ 0.01, *** *p* ≤ 0.001.

**Table 7 animals-08-00139-t007:** Behaviours of zoo mammals significantly affected in the three types of stratum use (terrestrial, arboreal, and scansorial) and by public category (small, medium and large).

Behaviour	Independent Variables	Estimate Coefficient (±SE)	*t* Values
Eating	Arboreal	0.22410 (±0.17278)	1.297
	Scansorial	−0.07479 (±0.14108)	−0.530
	Medium audience	0.17276 (±0.13783)	1.253
	Large audience	−0.12924 (±0.15546)	−0.831
	Arboreal*Medium	−0.17567 (±0.21897)	−0.802
	Scansorial*Medium	0.15114 (±0.19493)	0.775
	Arboreal*Large	0.77590 (±0.30631)	2.533 *
	Scansorial*Large	0.63908 (±0.21577)	2.962 **
Drinking	Arboreal	−0.01667 (±0.04938)	−0.338
	Scansorial	−0.01667 (±0.04032)	−0.413
	Medium audience	0.01126 (±0.03939)	0.286
	Large audience	0.19762 (±0.04443)	4.448
	Arboreal*Medium	−0.01126 (±0.06258)	−0.180
	Scansorial*Medium	0.01625 (±0.05571)	0.292
	Arboreal*Large	−0.19762 (±0.08754)	−2.257 *
	Scansorial*Large	−0.19762 (±0.06167)	−3.205 **
Inactive resting	Arboreal	0.4825 (±0.4685)	1.030
	Scansorial	0.7114 (±0.3826)	1.859
	Medium audience	0.9728 (±0.3738)	2.603
	Large audience	2.0656 (±0.4216)	4.900
	Arboreal*Medium	−0.5268 (±0.5938)	−0.887
	Scansorial*Medium	−0.9593 (±0.5286)	−1.815
	Arboreal*Large	−2.5856 (±0.8306)	−3.113 **
	Scansorial*Large	−2.4645 (±0.5851)	−4.212 ***
Alert	Arboreal	0.2000 (±0.1345)	1.487
	Scansorial	0.0100 (±0.1099)	0.091
	Medium audience	0.6198 (±0.1073)	5.775
	Large audience	0.3952 (±0.1211)	3.265
	Arboreal*Medium	−0.6526 (±0.1705)	−3.828 ***
	Scansorial*Medium	−0.3840 (±0.1518)	−2.530 *
	Arboreal*Large	−0.09524 (±0.2385)	−0.399
	Scansorial*Large	−0.3695 (±0.1680)	−2.199 *
Positive interaction	Arboreal	0.770909 (±0.195075)	3.952
	Scansorial	0.044242 (±0.159278)	0.278
	Medium audience	0.045931 (±0.155616)	0.295
	Large audience	−0.009091 (±0.175516)	−0.052
	Arboreal*Medium	1.179913 (±0.247219)	4.773 ***
	Scansorial*Medium	0.063042 (±0.220074)	0.286
	Arboreal*Large	−0.770909 (±0.345830)	−2.229 *
	Scansorial*Large	0.184329 (±0.243611)	0.757
Move away	Arboreal	−0.32413 (±0.27709)	−1.170
	Scansorial	−0.14913 (±0.22625)	−0.659
	Medium audience	0.01992 (±0.22104)	0.090
	Large audience	−0.01579 (±0.24931)	−0.063
	Arboreal*Medium	0.13560 (±0.35116)	0.386
	Scansorial*Medium	0.30909 (±0.31260)	0.989
	Arboreal*Large	0.01579 (±0.49123)	0.032
	Scansorial*Large	1.21103 (±0.34604)	3.500 ***
Non-visible	Arboreal	−1.4183 (±0.5563)	−2.550
	Scansorial	−1.0722 (±0.4542)	−2.361
	Medium audience	−0.7119 (±0.4438)	−1.604
	Large audience	−1.3615 (±0.5005)	−2.720
	Arboreal*Medium	0.9193 (±0.7050)	1.304
	Scansorial*Medium	0.7836 (±0.6276)	1.249
	Arboreal*Large	1.5215 (±0.9862)	1.543
	Scansorial*Large	2.5718 (±0.6947)	3.702 ***

* *p* ≤ 0.05, ** *p* ≤ 0.01, *** *p* ≤ 0.001.

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
