# Peer review of "The Different Physical and Behavioural Characteristics of Zoo Mammals That Influence Their Response to Visitors"

_animals, 2018, doi:10.3390/ani8080139_

Round 1

Reviewer 1 Report

The authors have responded to my previous comments but have not actually changed any of those sections of the text. 

Line 272 refers to the standoff barrier but does not give any indication of even the minimum distance from the public.  

The information about the animals being housed at the zoo for at least a year before the study, and no evidence of habituation to the presence of visitors should be included.

The interpretation of cortisol assay results is complicated by considerable individual variation, the fact that a cortisol response is associated with some non-stress stimuli, and that some stress responses may not involve elevated cortisol levels.  These have contributed to an increasing dissatisfaction with the use of cortisol to measure stress levels, added to the fact that the collection of some types of samples (blood) will add to the animals’ stress levels. Recommending this type of measurement to be included in future studies requires some expansion. 

Author Response

The authors have responded to my previous comments but have not actually changed any of those sections of the text. 

Line 272 refers to the standoff barrier but does not give any indication of even the minimum distance from the public.  DONE

The information about the animals being housed at the zoo for at least a year before the study, and no evidence of habituation to the presence of visitors should be included. DONE

The interpretation of cortisol assay results is complicated by considerable individual variation, the fact that a cortisol response is associated with some non-stress stimuli, and that some stress responses may not involve elevated cortisol levels.  These have contributed to an increasing dissatisfaction with the use of cortisol to measure stress levels, added to the fact that the collection of some types of samples (blood) will add to the animals’ stress levels. Recommending this type of measurement to be included in future studies requires some expansion. DONE

Reviewer 2 Report

Thank you for the opportunity to re-review this manuscript. The authors did a good job addressing my concerns. 

I think the this paper is potentially publishable with some additional revisions.

The list of behaviors included for analysis requires more explanation. For each behavior, the authors should provide a justification as to why the behavior is considered relevant to welfare, and in which direction a change in behavior would indicate decline in welfare state. For example, is more resting better or worse for welfare and upon what literature is that assertion being made? Is it the author's belief that the same pattern of correlation between change in behavior pattern and welfare consequences exists for every species studied? Why or why not?  A justification for each behavior must be included in the methods section and then the results must be discussed in the context of the author's predictions. If strong arguments cannot be made for why a behavior is relevant to welfare and grounded predictions related to direction/amount of change cannot be provided, then the behavior should be removed from the analysis OR the arguments in the paper should just be about changes in behavioral patterns without extension to welfare consequences.

Why show results for "Other" when you provide no detail about that suite of behaviors?

Provide a table that summarizes the significant results for each characteristic x audience size interaction. Currently, the reader has to pick through lots of n.s. results to get an appreciation for what changes were actually measured and the direction of these changes. More explanation is needed when the direction of effect is different from small to medium crowds than it is from medium to large crowds.

The entire predictive section must be removed. There is not nearly enough evidence provided in this paper to support these types of speculations. Sample sizes are very small, particularly for some of the life history traits, and the analysis does not even come close to  addressing the potential cumulative effects of multiple traits as the "predictive model" suggests. For example, you have terrestrial mammals as of more concern than arboreal or scansorial mammals presumably based on these results: "Following an increase in
 audience size, terrestrial animals ate less, drank more, slept more, and were more alert than the other two categories of animals for factor." Eating less + drinking more + sleeping more + more alert = most negative impact?? I have no idea how you came to this conclusion because no credible explanation is provided.

Author Response

The list of behaviors included for analysis requires more explanation. For each behavior, the authors should provide a justification as to why the behavior is considered relevant to welfare, and in which direction a change in behavior would indicate decline in welfare state. For example, is more resting better or worse for welfare and upon what literature is that assertion being made? Is it the author's belief that the same pattern of correlation between change in behavior pattern and welfare consequences exists for every species studied? Why or why not?  A justification for each behavior must be included in the methods section and then the results must be discussed in the context of the author's predictions. If strong arguments cannot be made for why a behavior is relevant to welfare and grounded predictions related to direction/amount of change cannot be provided, then the behavior should be removed from the analysis OR the arguments in the paper should just be about changes in behavioral patterns without extension to welfare consequences. Information added in the discussion about our vision of this issue.

Why show results for "Other" when you provide no detail about that suite of behaviors? “Other” REMOVED

Provide a table that summarizes the significant results for each characteristic x audience size interaction. Currently, the reader has to pick through lots of n.s. results to get an appreciation for what changes were actually measured and the direction of these changes. DONE

More explanation is needed when the direction of effect is different from small to medium crowds than it is from medium to large crowds. Details provided.

The entire predictive section must be removed. There is not nearly enough evidence provided in this paper to support these types of speculations. Sample sizes are very small, particularly for some of the life history traits, and the analysis does not even come close to  addressing the potential cumulative effects of multiple traits as the "predictive model" suggests. For example, you have terrestrial mammals as of more concern than arboreal or scansorial mammals presumably based on these results: "Following an increase in
 audience size, terrestrial animals ate less, drank more, slept more, and were more alert than the other two categories of animals for factor." Eating less + drinking more + sleeping more + more alert = most negative impact?? I have no idea how you came to this conclusion because no credible explanation is provided. Information added in the conclusion about our point of view in the proposed study – we explain that the model is not definitive, especially given its sample size but presented as a first iteration that will be subject to modification as other studies become available.

Reviewer 3 Report

General Comments:

While some improvements were made to the study, more details on the methods and statistical analysis are still needed in order to properly evaluate the results.

Specific Comments:

Ln 39: This rational is unclear “because they were less accustomed to close visual contact with people.” It was stated that this was reworded but I do not see any changes.

Ln 119: You should specify the exact time of day rather than saying morning.

Ln 137: A detailed ethogram should be provided even if you ultimately want to lump together behaviors across species.

Ln 148: If the data are not normally distributed, then GLM models are not appropriate since they are parametric. More details on the statistical analysis are still needed as it is not clear what was done. 

Author Response

Specific Comments:

Ln 39: This rational is unclear “because they were less accustomed to close visual contact with people.” It was stated that this was reworded but I do not see any changes. DONE

Ln 119: You should specify the exact time of day rather than saying morning.

A Latin Square design was used, which means that each animal was observed at a different time each day, but always during the morning. This method was used to avoid time of day bias as explained in the data collection section.

Ln 137: A detailed ethogram should be provided even if you ultimately want to lump together behaviors across species. Information added as a footnote in Table 2.

Ln 148: If the data are not normally distributed, then GLM models are not appropriate since they are parametric. More details on the statistical analysis are still needed as it is not clear what was done. 

The use of some statistical tests are beyond the verification of normal distribution of the data. Requirements for each possible test should be verified and it was made in the present paper. “Parametric” term was removed from the text.